# Structural basis for membrane attack complex inhibition by CD59

Emma C. Couves[1], Scott Gardner[1], Tomas B. Voisin[1], Jasmine K. Bickel[1,2], Phillip J. Stansfeld [3], Edward W. Tate [2] & Doryen Bubeck [1] ✉

CD59 is an abundant immuno-regulatory receptor that protects human cells from damage during complement activation. Here we show how the receptor binds complement proteins C8 and C9 at the membrane to prevent insertion and polymerization of membrane attack complex (MAC) pores. We present cryo-electron microscopy structures of two inhibited MAC precursors known as C5b8 and C5b9. We discover that in both complexes, CD59 binds the pore-forming β-hairpins of C8 to form an intermolecular β-sheet that prevents membrane perforation. While bound to C8, CD59 deflects the cascading C9 β-hairpins, rerouting their trajectory into the membrane. Preventing insertion of C9 restricts structural transitions of subsequent monomers and indirectly halts MAC polymerization. We combine our structural data with cellular assays and molecular dynamics simulations to explain how the membrane environment impacts the dual roles of CD59 in controlling pore formation of MAC, and as a target of bacterial virulence factors which hijack CD59 to lyse human cells.

The membrane attack complex (MAC) is a human immune pore that ruptures lipid bilayers to kill cells. While a potent weapon of innate immune defense, MAC assembled on human cells has devastating consequences for disease pathologies[1,2]. A critical regulator of MAC is CD59, a GPI-anchored cell surface receptor. CD59 is the only membrane-bound inhibitor of MAC on human cells and is the last line of defense against activation of the complement terminal pathway[3]. Without it, MAC pores formed on red blood cells result in rupture, causing chronic hemolysis and immune-related neuropathies[4]. CD59 is also highly expressed in B-cell lymphomas, contributing to immune evasion and protection from antibody-based therapeutics that activate complement[5,6]. Therefore, understanding the molecular basis by which CD59 inhibits MAC is essential for our understanding of immune regulation in health and disease.

CD59 prevents MAC membrane perforation and polymerization on the surface of human cells. MAC assembles from the stepwise association of five complement proteins: C5b, C6, C7, C8, and C9[7] (Fig. 1). Though unable to bind soluble C8 or C9, CD59 directly engages both proteins within the nascent MAC to trap C5b8 and C5b9 assembly intermediates[8]. Mutagenesis experiments and antibody mapping data have identified several discontinuous clusters of CD59 residues that are important for activity[9,10]. However, CD59 binds more than one complement protein within the context of two assembly intermediates. It is therefore challenging to attribute specific interaction interfaces without structural information for the complex.

C8 and C9 both bind CD59 through their pore-forming membrane attack complex perforin (MACPF) domains[9,11]. The pore-forming machinery of complement proteins is encoded within a structurally conserved domain shared by other immune pores[12] as well as bacterial virulence factors from the cholesterol-dependent cytolysins (CDCs) superfamily[13]. During pore formation, a pair of helical bundles within the MACPF undergoes dramatic secondary structure rearrangements to form two transmembrane β-hairpins (TMH1 and THM2)[14] which arrange into a giant β-barrel pore[15]. Although structures of C8 and C9 in both soluble and membrane-inserted conformations have been

[1]Department of Life Sciences, Sir Ernst Chain Building, Imperial College London, London SW7 2AZ, United Kingdom. [2]Department of Chemistry, Molecular Sciences Research Hub, Imperial College London, London W12 0BZ, United Kingdom. [3]School of Life Sciences and Department of Chemistry, Gibbet Hill Campus, The University of Warwick, Coventry CV4 7AL, United Kingdom. ✉e-mail: d.bubeck@imperial.ac.uk

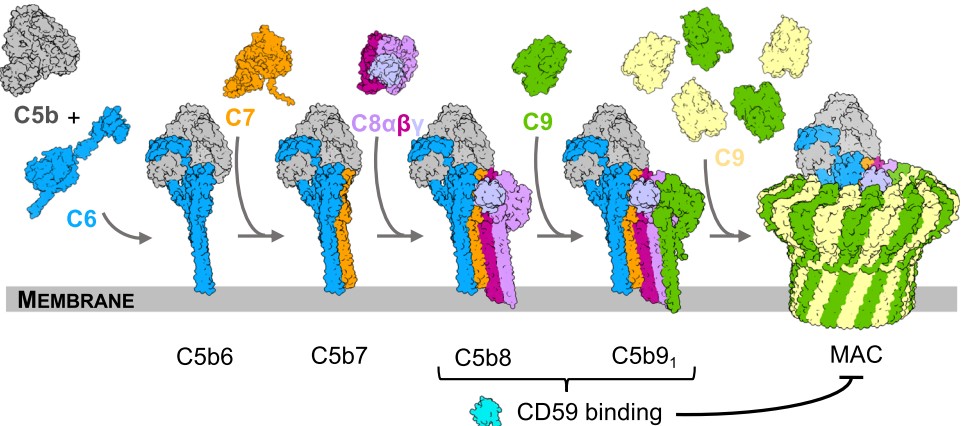

**Fig. 1 | Schematic of MAC assembly.** The MAC pore is formed from the sequential and stepwise assembly of complement proteins: C5b, grey; C6, blue; C7, orange; C8α, pink; C8β, magenta; C8γ, light purple; C9, alternating monomers are yellow and green. During assembly, complement proteins undergo dramatic structural rearrangements in which two helical bundles within their MACPF domains unfurl into membrane-inserting β-hairpins. CD59 (cyan) binds at two stages of this assembly process (C5b8 and C5b9) to block membrane perforation and C9 polymerization. Images are rendered from structural models. C5b6 and all MAC assemblies: PDB ID: 6H03; Soluble forms of complement proteins are derived from C6: PDB ID: 3T5O; C8: PDB ID: 3OJY; C9: PDB ID: 6CXO. C7 was derived from an AlphaFold2 prediction: AlphaFold Protein Structure Database P10643.

solved[15–18], how CD59 captures these proteins to prevent pore formation remains unresolved.

Current models for how CD59 engages complement proteins are based largely on structural principles derived from complexes with bacterial pore-forming proteins that hijack CD59 to target human cells for lysis[19]. Despite a strong correspondence of residues within CD59 that bind CDCs, C8 and C9[20], CD59 does not engage the pore-forming machinery of CDCs. Instead, these CDCs recruit CD59 through their membrane-binding domain to facilitate oligomerization and augment pore formation[21]. Given these dramatic differences in end outcomes, it remains unclear how CD59 binding contributes to MAC pore inhibition.

In this study, we set out to understand how CD59 protects human cells from MAC damage. We use cryo-electron microscopy (cryoEM) to determine the structures of both CD59-inhibited MAC complexes (C5b8-CD59 and C5b9-CD59) within the context of a lipid nanodisc. We show how a single CD59 simultaneously inhibits C8 and C9 by altering the trajectories of pore-forming β-hairpins. Additionally, we discover that disrupting membrane insertion of C9 indirectly halts MAC polymerization. By combining our structural data with molecular dynamics (MD) simulations, we reveal that the orientation of CD59 relative to the membrane defines the specificity of its mode of action in immune regulation and host-pathogen interactions.

## Results

CD59 can act on both C5b8 and C5b9 MAC precursors to stop membrane perforation and polymerization of C9[8,22,23]. To understand how MAC is blocked on human cells to prevent disease, we captured inhibited complexes on CD59-decorated lipid nanodiscs and visualized both C5b8-CD59 and C5b9-CD59 complexes by cryoEM. The extracellular domain of CD59 was expressed in bacteria, refolded, and conjugated to a myristoylated cytotopic peptide to mimic the natural GPI anchor[24]. Peptide-modified CD59 was then embedded in DOPC lipid nanodiscs, and individual complement proteins (C5b6, C7, C8, and C9) were sequentially added to CD59-decorated nanodiscs to form either C5b8-CD59 or C5b9-CD59 assemblies. Complexes were purified through sucrose density centrifugation and GraFix (gradient fixation) with glutaraldehyde[25] applied to prevent denaturation of complexes when preparing cryoEM grids (Supplementary Fig. 1).

### CryoEM structure of the C5b8-CD59 complex

C8 is a heterotrimeric protein composed of C8α, C8β, and C8γ polypeptide chains[26]. While C8β is responsible for binding the MAC

assembly precursor, C8α is the first MAC protein to penetrate the lipid bilayer[27]. To understand how CD59 blocks membrane insertion of C8α we solved the structure of the C5b8-CD59 complex in a lipid nanodisc. Using 3D classification of images we resolved an equimolar complex consisting of a single copy of C5b, C6, C7, C8, and CD59 (Supplementary Fig. 2). Although the global resolution was 3.0 Å, the local resolution of the density corresponding to CD59 was initially poor and prevented modeling (Fig. 2A and Supplementary Fig. 3A). To improve the resolution of the C8-CD59 interface we subtracted density corresponding to C5b, C6, and C8γ and focused our alignment on the region of the map composed of C7, C8β, C8α, and CD59, enabling us to resolve the central β-sheet of CD59 and unambiguously define its orientation within the map (Fig. 2B, C and Supplementary Fig. 3B). Initial models for complement proteins were built using coordinates for the soluble MAC structure (sMAC)[28] and the CD59 crystal structure[29]. Models were refined using adaptive distance restraints with iterative building and refinement of side chains where density permitted. This allowed us to build a near-complete atomic model for the C5b8-CD59 inhibited complex (Fig. 2B; Supplementary Table 1).

### CD59 captures C8α β-hairpins

During MAC formation helical bundles within the MACPF domains of complement proteins unfurl to form two membrane-associating or transmembrane β-hairpins (TMH1 and TMH2)[30]. In the C5b8 inhibited complex, helical bundles of C6, C7, C8α, and C8β are fully extended, with CD59 directly binding the leading β-strand of C8α TMH2 (Fig. 3A). While the curvature of the first two β-strands (TMH1) of C8α is templated by C8β, the geometry of the second β-hairpin (TMH2) is influenced by CD59. CD59 binding bends the trajectory of C8α TMH2 30° from its membrane-inserted conformation, thus preventing the tip of the hairpin from puncturing the lipid bilayer.

Within the C8α-CD59 interface, the C-terminal β-strand of CD59 forms a contiguous antiparallel β-sheet with the MACPF domain of C8α (Fig. 3B). The interaction is initiated by a salt bridge between CD59:E58 and C8α:K376, which is located on the leading edge of TMH2. Next, three consecutive glycine residues on C8α (C8α:G375, C8α:G374, C8α:G373) encode flexibility in the strand curvature and enable the β-strand bend observed in our map (Fig. 3A, Supplementary Movie 1). Though not modeled in our maps, there is an additional stretch of 3 glycine residues on the leading strand of TMH2 (C8α:363, C8α:362, C8α:361) that could further contribute to strand flexibility near the membrane. Aromatic residues of CD59 (CD59:F47 and CD59:Y61) fill

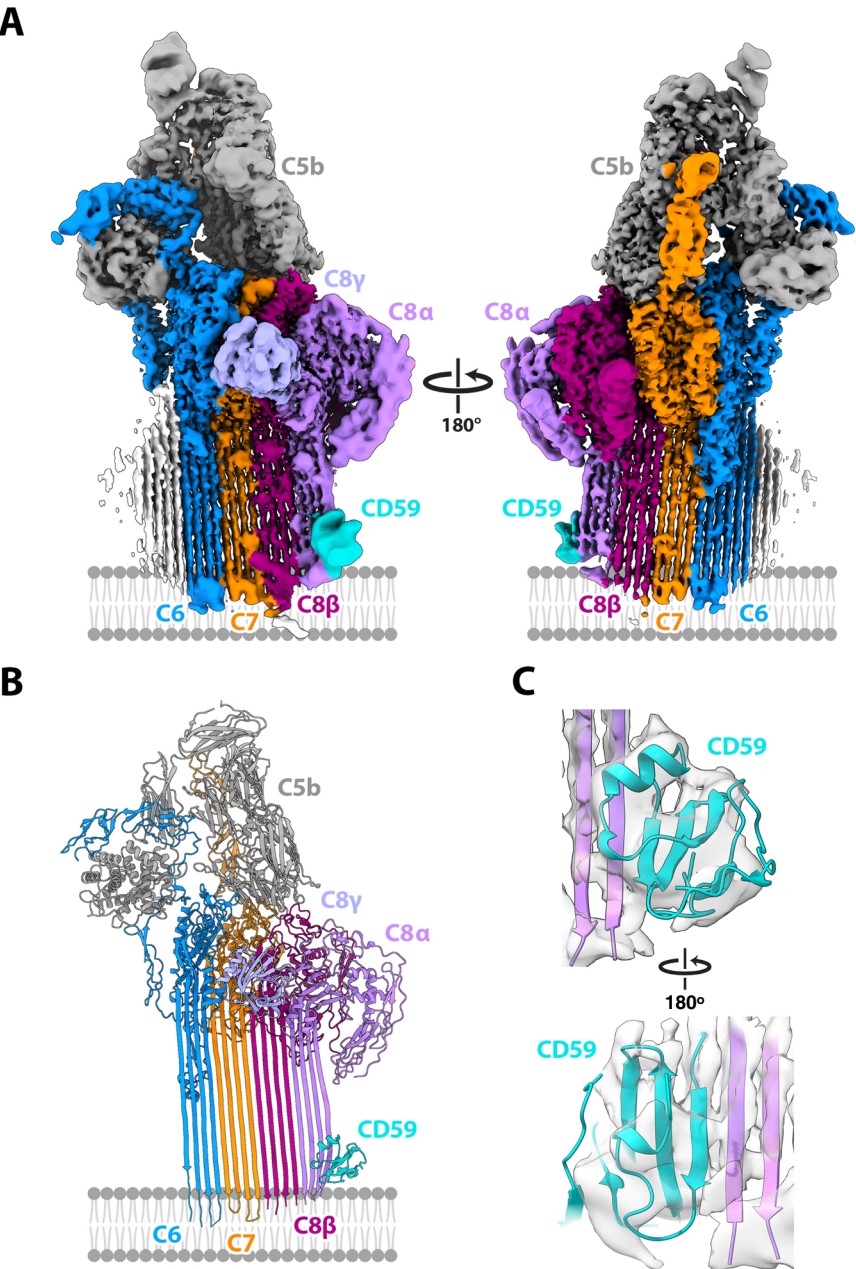

**Fig. 2 | Structure of the C5b8-CD59 complex.** CryoEM map **A** and model **B** for the C5b8-CD59 nanodisc complex. Complement proteins are colored (C5b, grey; C6, blue; C7, orange; C8α, pink; C8β, magenta; C8γ light purple). CD59 is cyan. The membrane, not visible in the sharpened map, is schematized for reference. To improve the resolution for CD59, the density for C5b, C6 and C8γ was subtracted from the consensus map (**A**) and refined. **C** Model for CD59 and interaction interface with C8α overlayed on the density subtracted focus refined map.

the void created by the first C8α glycine bend (Fig. 3A). Previous mutagenesis studies altering these aromatic residues abrogate CD59 activity[9,20,31], with substitution of CD59:Y61 for a glycine resulting in particularly marked failure to inhibit MAC-mediated cell lysis[9]. Taken together, these data suggest that aromatic side chains of CD59 at the C8α interface inhibit MAC by reshaping the trajectory of TMH2 towards the membrane.

The C8α strand trajectory is further stabilized by a network of interactions across the C8α-CD59 interface. Backbone atoms of the glycine bend contribute to a series of hydrogen bonds that stabilize the contiguous β-sheet formed between C8α and CD59 (C8α:G375-CD59:L59; C8α:G374-CD59:T60; C8α:G373-CD59:Y61; C8α:K371-CD59:Y62) (Fig. 3B). The intermolecular β-sheet is further stabilized by side-chain interactions formed across adjacent β-strands (CD59:Y61-C8α:K371; CD59:Y62-C8α:K370; CD59:Y62-C8α:F372). These structural data explain previous mutagenesis studies showing that both Y61 and Y62 of CD59 are essential for inhibiting MAC-mediated cell death[9,31].

CD59 orthologs from non-primate species have limited effect in protecting cells from human MAC[32]. While the CD59-binding site is highly conserved across primates, sequences diverge for other species with the largest differences observed in rodents (Supplementary Fig. 4A–B). To understand the molecular basis underpinning CD59 species specificity, we compared the CD59-C8α binding site in our structure with models generated from murine sequences. In the human structure, CD59:E58 forms a salt bridge with K376 on the leading edge of C8α TMH2 (Supplementary Fig. 4C). While this salt bridge is conserved in the murine model (CD59:K58 and C8α:D362), the positions of the charged residues are reversed (Supplementary

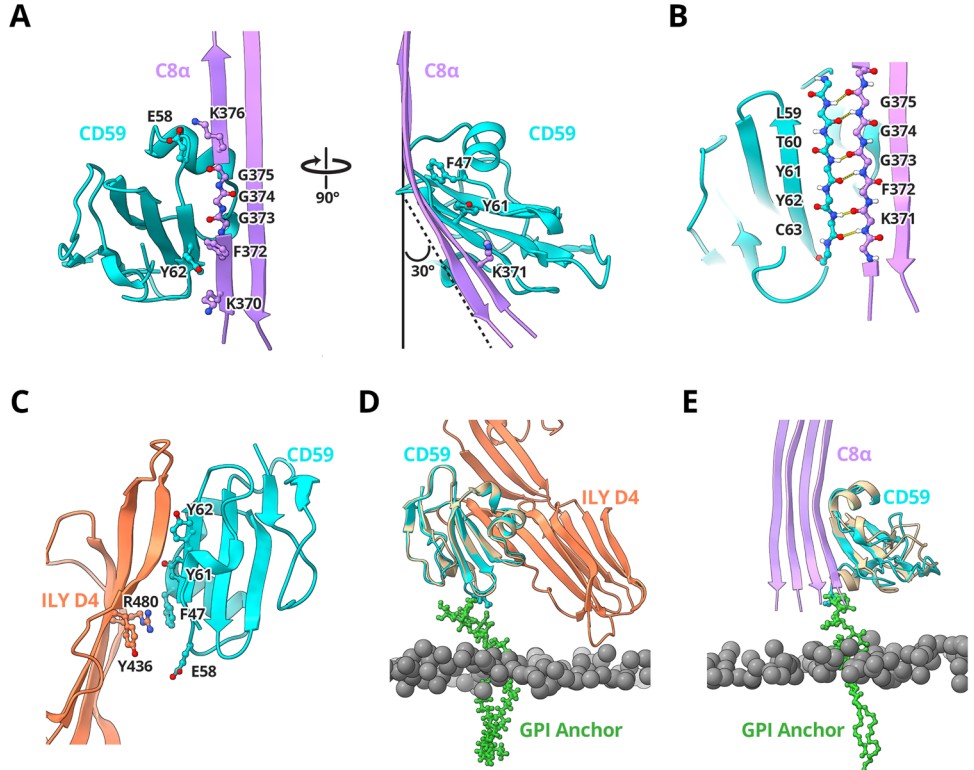

**Fig. 3 | CD59 interaction interfaces. A** Ribbon diagram of the C8α-CD59 interface. CD59 (cyan) captures the extending TMH2 residues of C8α (pink). Aromatic residue CD59:F47 binds C8α glycines (G373, G374, G375) to bend the β-hairpin trajectory. The C8α-CD59 interface is further stabilized by a salt bridge between CD59:E58 and C8α:K376, together with a pair of consecutive tyrosine-lysine interactions on either side of the β-sheet (CD59:Y62-C8α:K370; CD59:Y61-C8α:K371). Key residues that mediate the interactions are shown as sticks. **B** Hydrogen-bonding pattern of backbone atoms within the intermolecular antiparallel β-sheet. **C** Ribbon diagram of the ILY-CD59 interaction interface (PDB ID: 5IMT)[33]. ILY (orange) binds CD59 (cyan) through a β-hairpin extension of domain 4 (D4). The ILY-CD59 interface is comprised of an intermolecular anti-parallel β-sheet, including CD59 residues that engage C8α. **D** Superposition of the ILY-CD59 crystal structure (PDB ID: 5IMT) with a pose from the atomistic molecular dynamics simulation of GPI-anchored CD59 (tan). **E** Superposition of the C5b8-CD59 structure with a different pose from the atomistic molecular dynamics simulation of GPI-anchored CD59 (tan). In this position, CD59 is rotated 106° relative to its position in (**D**). Phosphorous atoms from lipid headgroups are grey spheres, simulated GPI anchor for CD59 is shown in green sticks. Initial and final configurations for the three MD repeats are included in the Supplementary Data Files.

Fig. 4D). CD59:Y62 plays an important role in controlling MAC susceptibility in humans; however, the equivalent position in the murine model is an arginine residue (CD59:R62). This change is compensated by concurrent charge-complementary mutations in C8α (S358; E356). Taken together, our analysis shows how CD59 and C8α have co-evolved to provide immune protection that drives the species selectivity of complement.

### Comparison with the CDC binding site

CD59 residues within the C8α-binding site are also involved in binding some cholesterol-dependent cytolysins[19]. Although the pore-forming machinery of MAC and CDCs are highly conserved, the end outcomes of CD59 binding are diametrically opposed. Intermedilysin, lectinolysin and vaginolysin are CDCs that require CD59 for membrane-binding and polymerization. All three bind CD59 through an extended β-tongue within the membrane recognition domain (domain 4, D4)[33], hijacking the C8α binding site on CD59 and repurposing it to target host cells (Fig. 3A, C). While CD59 residues CD59:F47, CD59:Y61, CD59:Y62 contribute to both MAC and CDC binding interfaces[20], they do so in different ways. In contrast to its role in defining the C8α glycine bend in the inhibited MAC complex, CD59:F47 stabilizes the interface with CDCs vaginolysin and intermedilysin through aromatic ring stacking (Fig. 3C)[21,33]. Similarly, the antiparallel β-sheet of CD59 is continued in both cases either by the pore-forming domain of C8α or the membrane-binding domain of CDCs; however, the most striking difference between complexes with MAC and CDCs is that CD59 is

rotated 106° with respect to the membrane (Fig. 3D, E). We used atomistic molecular dynamics simulations of the full-length receptor in a lipid bilayer to understand how a GPI-anchored CD59 could accommodate both binding modes and observed a wide range of orientations of CD59 relative to the membrane, including those in the CDC-CD59 and C5b8-CD59 structures (Fig. 3D, E; Supplementary Fig. 5).

### CryoEM structure of the C5b9-CD59 complex

In addition to preventing membrane insertion of C8α, CD59 also blocks C9 membrane rupture and polymerization[22]. To understand how CD59 acts on both MAC assembly intermediates, we next used cryoEM to solve the structure of the membrane-bound C5b9-CD59 complex (Supplementary Table 1). Unlike the single stoichiometry observed for the C5b8-inhibited complex, our cryoEM analysis reveals that C5b9-CD59 is a heterogenous assembly that varies in the extent of C9 polymerization (Fig. 4 and Supplementary Fig. 6). Using 3D classification to separate homogeneous populations of particles, we find that CD59 inhibits C5b9 with either 2, 3 or 4 copies of C9. Maps were refined to a global resolution of 3.3 Å for C5b9$_2$-CD59, 3.3 Å for C5b9$_3$-CD59 and 3.7 Å for C5b9$_4$-CD59 (Supplementary Fig. 3C–F). The density for CD59 within each of these maps was further improved by density subtraction and focused refinement strategies analogous to those used in the C5b8-CD59 structure (Supplementary Fig. 7A–D).

A comparison of the C5b8-CD59 and C5b9-CD59 structures reveals a common mechanism whereby CD59 captures the extending β-hairpins of C8α and prevents them from crossing the lipid bilayer

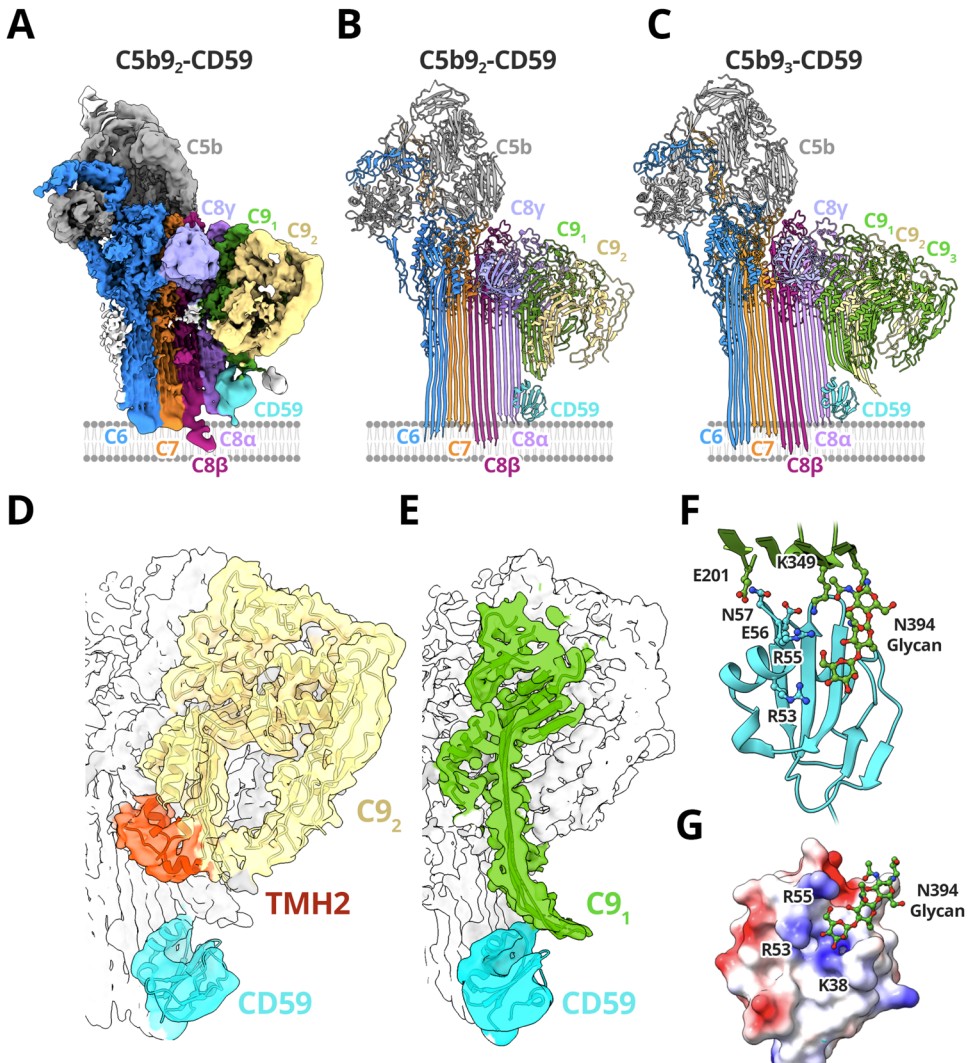

**Fig. 4 | Structure of the C5b9-CD59 complex.** CryoEM map **A** and model **B** of C5b9-CD59 complexes comprised of 2 molecules of C9 (C9$_1$ and C9$_2$). (**C**) A model for the C5b9-CD59 complex comprised of 3 C9 molecules. Complement proteins are colored (C5b, grey; C6, blue; C7, orange; C8α pink, C8β magenta, C8γ light purple); CD59 is cyan. The membrane, not visible in the sharpened map, is schematized for reference. **D**, **E** Density for the C5b9$_2$-CD59 map after density subtraction and focused refinement (transparent surface) overlayed with the model. **D** For the terminal C9 (C9$_2$, yellow), only the first transmembrane β-hairpin has unfurled while TMH2 (red) remains helical. **E** CD59 (cyan) deflects both TMH1 and TMH2 from the first C9 (C9$_1$, green) to block membrane insertion. **F** The C9$_1$-CD59 interaction interface. The key residues that mediate interactions are shown as sticks. **G** Electrostatic surface representation of CD59 highlighting positively charged residues (R55, R53, K38) that interact with the C9 glycan extending from N394 (green sticks). Coulombic potential calculated for the CD59 model in B ranging from −10 (red) to +10 (blue) kcal/(mol·e).

(Figs. 2, 4A–C). While the β-strands of TMH2 still deviate from their MAC trajectory, the angle is less pronounced in the C5b9-inhibited complexes. We, therefore, attribute these differences to constraints in β-barrel curvature imposed by additional C9 MACPFs and by contacts of C9 with CD59. Indeed, C9 continues to incorporate into the MAC precursor with up to four copies. Notably, we do not observe complexes with a single C9 (Supplementary Fig. 6).

## CD59-binding indirectly blocks MAC polymerization

Within the C5b9-CD59 complex, C9 MACPF domains undergo dramatic conformational rearrangements in the inhibited complex. We find that both TMH1 and TMH2 residues within the first C9 (C9$_1$) completely transition from their helical to hairpin states (Fig. 4A–C). By contrast, the terminal C9 is in a transitory state for complexes with two or three molecules of C9. Here, only TMH1 has unfurled while TMH2 remains helical (Fig. 4D), similar to the terminal C9 of the soluble inhibited MAC (sMAC) complex[28]. As in sMAC, the helical conformation of TMH2 sterically blocks further polymerization of C9. While our structural analysis shows some complexes with weak density for a fourth C9 molecule, this final monomer is not well resolved and may comprise a soluble C9 that is transiently associated with the complex. Indeed, a conformational landscape analysis of the C5b9$_3$-CD59 complex showed that for some particles, the MACPF of terminal C9 is not aligned with that of the preceding monomer and may represent the initial engagement with soluble C9 (Supplementary Fig. 7E). Together, our structural data support a model whereby CD59 indirectly blocks MAC polymerization by trapping a transitory state of C9 that prevents additional monomers from stably binding.

## CD59 deflects C9 β-hairpins

Although transmembrane residues of C9$_1$ have unfurled into their β-hairpin conformation, CD59 prevents these residues from reaching the lipid bilayer. Bound to C8α, CD59 is positioned directly below the path of the extending β-hairpins of C9$_1$. A comparison with the MAC conformation of C9 shows that CD59 deflects the extending β-hairpins 60° from their transmembrane trajectory (Fig. 4E). Indeed, a glycine

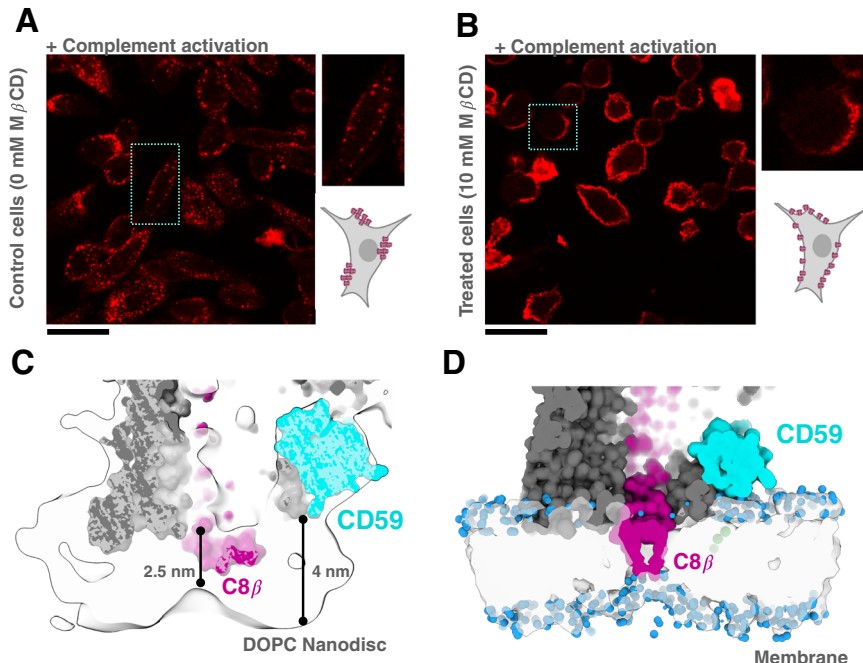

**Fig. 5 | Influence of the membrane environment on MAC assembly and inhibition. A, B** Cholesterol depletion assays. A representative image (out of 10 randomly selected locations) for each condition is shown. Scale bars, 50 μm. **A** Complement was activated on CHO cells with a polyclonal anti-CHO IgG antibody. Cells were incubated with C9-depleted human serum supplemented with a chemically labeled fluorescent C9 (C9-Alexafluor 568) capable of forming MAC. **B** CHO cells treated with MβCD to deplete cholesterol. Complement activation and C9 detection is as described in (**A**). Insets show a zoomed in view of single cell.

Cartoon schematics highlight the pattern of MAC deposition. **C** CryoEM map of the C5b9₂-CD59 complex in a lipid nanodisc applying a positive B-factor of +50 Å² (transparent surface). Surface rendering of the protein model is underlayed. CD59 is cyan, C8β is magenta and the remaining complement proteins are grey. **D** Map generated from the coarse-grained model of the C5b8-CD59 complex including the GPI anchor (green). Protein components colored as in panel (**C**). Water molecules in proximity to the membrane are shown as blue spheres. Initial and final configurations for the three MD repeats are included in the Supplementary Data Files.

residue (C9:G350) lies within a stretch of TMH2 exhibiting the largest curvature. Our structures show how a single CD59 molecule can simultaneously engage C8α and deflect the cascading β-hairpins of C9 to protect human cells from damage.

The interface between the C9 β-hairpins and CD59 is stabilized largely by electrostatic interactions. Residues within two loops of CD59 (residues 55–57 and 32–34) form a series of salt bridges and hydrogen bonds with the extending C9₁ TMH2 residues (Fig. 4F). Our structural data reconcile seemingly contradictory studies mapping the MAC-binding site to residues CD59:53-57[10] and residues CD59:27–38[34]. Specifically, antibody mapping studies previously identified CD59:E56 as central to the MAC-binding site[10]. In our structure CD59:E56 forms a salt-bridge with C9:K349, consistent with mutagenesis data showing that a charge reversal at this position (E56R) disrupts CD59 activity[10]. Similarly, charge reversal substitutions R53E and K38E reduced CD59 activity[10]. In the C5b9-CD59 complex, these residues form part of a pocket of positively charged CD59 residues (R55, R53, and K38) near a glycan extending from the TMH2 strand (C9:N394) (Fig. 4G). Strikingly, within the sMAC structure this glycan coordinates a conformation of C9 involved in templating β-hairpins of the polymerizing MAC[28]. In summary, our data define the CD59-C9 interaction interface and suggest a mechanism for how CD59 might interfere with glycan-mediated templating of C9 β-hairpins.

### Role of lipids in CD59 activity

CD59 is enriched in cholesterol-containing lipid microdomains of the plasma membrane[35], consistent with our data showing clusters of CD59 expressed on the surface of CHO cells (Supplementary Fig. 8A). Given that pore formation is independent of cholesterol in vitro[15,30], we tested whether cholesterol depletion alters the distribution of MAC on cells. We activated complement on CHO cells using a polyclonal IgG antibody and supplemented C9-depleted human serum with a

chemically labeled fluorescent C9 capable of forming MAC (Supplementary Fig. 8). We observed that during complement activation, MAC is assembled in punctate clusters on the cell surface analogous to those observed for CD59 (Fig. 5A and Supplementary Fig. 8A). Although MAC was able to form on cholesterol-depleted cells, pores were more diffusely distributed (Fig. 5B). Together, our data are consistent with a model whereby upstream complement activation pathways cluster MAC assembly in cholesterol-containing microdomains that are enriched with CD59. These data are consistent with assays showing monoclonal antibody therapeutics that activate complement target membrane rafts[36].

Our structures show that CD59 captures the extending β-hairpins of complement proteins as pores assemble on the membrane. With distributions of CD59 and MAC dependent on cholesterol, we analyzed how MAC assembly affects the local lipid environment. By applying a positive B-factor to our C5b9-CD59 map, we discover that the membrane-anchored β-hairpins of C8β cause a local thinning of the bilayer and create a 'pinch point' in the membrane (Fig. 5C). We next created a coarse-grained model of our C5b8-CD59 complex that included the GPI anchor (Supplementary Fig. 9) and ran MD simulations within a lipid membrane. Our simulations show that the lipidic edge near C8β thins yet remains impermeable to water (Fig. 5D), consistent with the density map. Together, these structural and computational data show how complement proteins locally distort the lipid bilayer to alter physical properties of cellular membranes[30].

### Discussion

CD59 is the final safeguard in an irreversible cascade in which reaction kinetics are a driving factor. MAC assembly is initiated by the proteolytic cleavage of C5 by the C5 convertase. The newly formed C5b is highly labile and without the rapid association of C6, decays to an inactive conformation unable to nucleate MAC[37]. Reaction rates

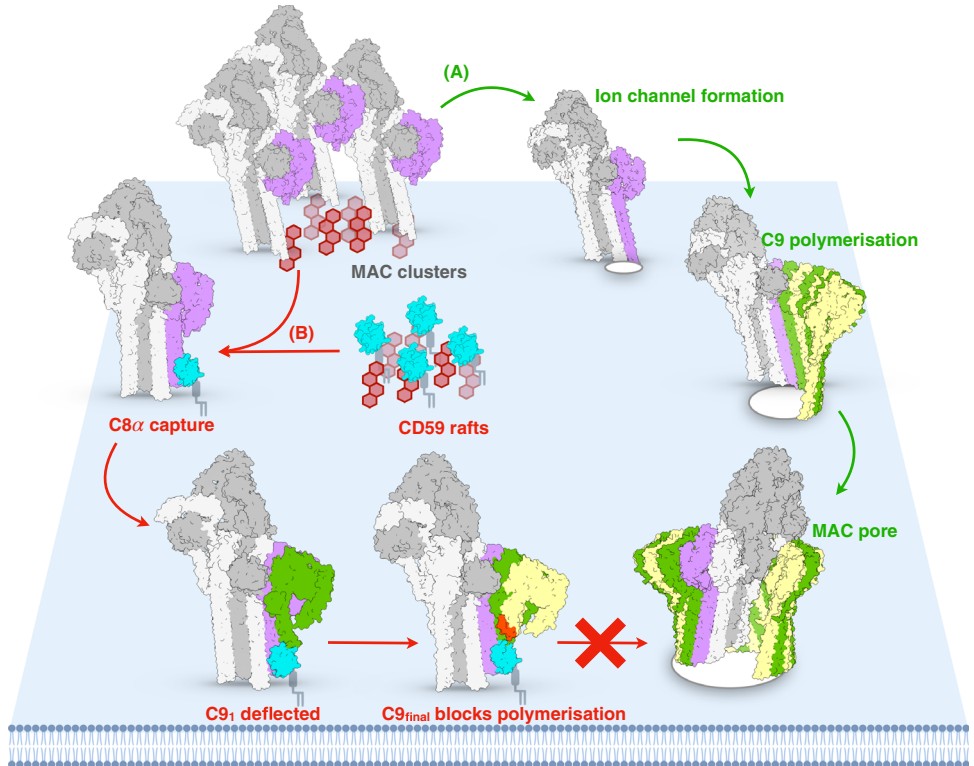

**Fig. 6 | Schematic of MAC assembly and its inhibition by CD59. A** MAC assembles on cholesterol-rich microdomains (red hexagons) in the plasma membrane. While C7 anchors the growing MAC, C8β thins the bilayer and primes the membrane for rupture by C8α (pink). C9 joins the assembly, undergoing discrete structural transitions to form the pore. The MACPF domain of soluble C9 makes an initial contact and aligns the central β-sheet. The pore-forming β-hairpins extend sequentially. TMH1 is followed by TMH2, which allow C9 polymerization to complete the MAC pore. **B** CD59 is a GPI-anchored cell surface receptor that clusters in cholesterol-rich microdomains. Upon complement activation, CD59 could respond to membrane thinning by C8β and capture C8α as its transmembrane β-hairpins are extending (pink). While bound to C8α, CD59 is positioned to deflect the cascading C9 β-hairpins and divert their membrane trajectory (green). The next C9 molecule (yellow) is able to bind but trapped in an intermediate conformation in which TMH2 remains helical (red) and in which C9 polymerization is halted.

also tip the balance in the final step of MAC formation, with the addition of the first C9 molecule serving as the kinetic bottleneck in C9 polymerization[38]. Our structures show that CD59 catches the extending transmembrane β-hairpins of C8α and bends their trajectory to prevent membrane penetration. Once bound, CD59 is then positioned to deflect the cascading transmembrane β-hairpins of C9 (Fig. 6). Although CD59 only directly contacts the first C9, inhibited complexes can incorporate up to three additional C9 monomers. Given that polymerization of C9 occurs at a faster rate than starting new pores, CD59 likely leverages assembly kinetics by locally reducing the concentration of C9. Furthermore, our data suggest that CD59 is recruited to the assembling MAC prior to C8α-MACPF structural transitions. We, therefore, speculate that the rate-limiting step in MAC assembly may be encoded within C8, the only MAC component with two MACPF domains.

In addition to direct protein interactions, we propose that the lipid environment plays an important role in MAC regulation (Fig. 6). Our cellular data show that MAC assembly is clustered on the plasma membrane and that this distribution depends on cholesterol-rich microdomains. Initiation of the complement terminal pathway is determined by local assembly of the C5 convertase. Therefore, complement deposition on self-cells may be preferentially activated in lipid rafts enriched in GPI-anchored CD59. Structural changes within the membrane during MAC assembly could further influence CD59 recruitment. It was previously observed that complement proteins alter biophysical properties of the lipid bilayer even before the pore is formed[30]. Here we demonstrate that the transmembrane hairpins of C8β are fully extended and locally thin the membrane to create a

'pinch point'. While the implications of this thinning on MAC function remains to be explored, hydrophobic mismatches between proteins and the lipid bilayer can influence membrane protein interactions and diffusion[39]. CD59 may be recruited to sites of MAC assembly by responding to lateral raft-lipid interactions within cholesterol-containing microdomains, in agreement with single-molecule studies showing CD59 responds to changes in transbilayer raft phases[40].

Though a potent complement regulator, CD59 is also exploited by pathogens that target human cells. In the case of bacterial virulence factors, some cholesterol-dependent cytolysins co-opt CD59 as a cell receptor to facilitate pore formation[41]. Similar to the C8α-CD59 interface, CD59 binds these CDCs through its outer β-strand to form an intermolecular antiparallel β-sheet[21]. However, unlike its deflection of complement pore-forming hairpins, CD59 coordinates CDC-membrane-binding domain oligomerization[42]. CDC oligomers then undergo specific structural transitions that collapse the complex towards the membrane and allow transmembrane β-hairpins to rupture the bilayer. Intriguingly, we have found that the orientation of CD59 relative to the membrane differs dramatically between its pore-inhibiting and pore-augmenting activities. Computational models for GPI-anchored proteins reveal that the highly flexible phosphoethanolamine linker can drastically change the orientation of the extracellular domain[43]. Our atomistic MD simulations of GPI-anchored CD59 support this model and show a wide range of flexibility relative to the membrane.

Despite these differences, complement proteins and pathogens may share a common mechanism of cholesterol-mediated CD59 recruitment. CDCs directly bind cholesterol-rich membranes through

loops that anchor into the bilayer[44]. MD simulations show that cholesterol-binding loops of CDCs locally order lipids[45]. For those CDCs that hijack CD59, membrane anchoring by these loops may mimic biophysical changes in the membrane caused by early stages of MAC assembly. The bacterium *Pseudomonas aeruginosa* also exploits CD59 and the endocytic machinery of flotillins to promote pathogen invasion[46]. Finally, several enveloped viruses target membranes enriched in CD59 for budding, thus providing a mechanism for complement immune evasion[47,48].

CD59 is the only membrane-bound regulator of MAC. Together our structural, cellular, and computational experiments provide insight into an essential mechanism of immune regulation. Our structures explain how CD59 prevents membrane insertion of pore-forming β-hairpins and further polymerization of MAC. In conclusion, our data suggest a model in which lipids may transduce immune activation signals that can be hijacked by pathogens that target human cells.

## Methods

### C5b8-CD59 and C5b9-CD59 sample preparation

MSP2N2-His$_6$ protein[49] was expressed by transforming *E. coli* BL21 DE3 cells with pET28-MSP2N2 (Addgene). Cells were grown to a OD$_{600}$ of ~0.8 in the presence of 50 μg/mL of kanamycin and induced for 3 hours with 0.5 mM isopropy β-D-1-thiogalactopyranoside (IPTG). Cells were pelleted by centrifugation and resuspended in lysis buffer (40 mM Tris-HCl pH 8.0, 0.3 M NaCl, cOmplete protease inhibitor (Roche), 5 μg/mL DNase I, 1% v/v Triton X-100). Cells were lysed by sonication and then purified via a TALON cobalt affinity resin (Clontech). The resin was washed with 40 mM Tris-HCl, pH 8.0, 0.3 M NaCl 0.5% v/v Triton X-100, followed by 40 mM Tris-HCl, pH 8.0, 0.3 M NaCl 50 mM sodium cholate. The protein was eluted from the column with 40 mM Tris-HCl, pH 8.0, 0.3 M NaCl, 0.3 M imidazole, then purified by size exclusion chromatography using a Superdex 200 10/300 column (GE healthcare) run in 40 mM Tris-HCl, pH 8.0, 0.1 M NaCl, 0.5 mM EDTA.

Nanodiscs were prepared by resolubilising 1.88 mg of 1,2-dioleoyl-sn-glycero-3-phosphocholine (DOPC) (Avanti Polar Lipid) in 40 mM Tris-HCl, 0.1 M NaCl, 0.5 mM EDTA, 64 mM sodium cholate and mixing in a 1:100 (MSP2N2:DOPC) molar ratio with purified MSP2N2. The mixture was incubated on ice for 20 minutes before the addition of preactivated Bio-beads SM2 (Bio-Rad) overnight at 4 °C agitated. The mixture was then purified over a Superose 6 10/300 column (GE healthcare) equilibrated in 40 mM Tris-HCl, pH 8.0, 0.1 M NaCl, 0.5 mM EDTA. Fractions containing 17 nm nanodiscs were pooled and concentrated.

To generate CD59-inihibited complement complexes for structural studies we used a recombinantly expressed version of the soluble domain of CD59 modified with a C-terminal cytotopic peptide for membrane anchoring (Richard Smith, King's College London)[50]. Preformed DOPC nanodiscs were incubated with this modified CD59 before the addition of C5b6, C7, C8, and C9 (for the c5b9 complex) (CompTech). Individual complement proteins were added sequentially with 5 minute incubation at 37 °C such that the final molar ratio was 10:1:1:1:2 (CD59:C5b6:C7:C8:C9). For the C5b8-CD59 sample, C9 was omitted from the reaction. Samples were subjected to purification by density centrifugation. Continuous gradients of 10-30% sucrose containing 0.15% glutaraldehyde, 120 mM NaCl, and 20 mM HEPES pH 7.5 were centrifuged at 204384.5 x *g* using a SW60-Ti rotor for 18 h at 4 °C with minimum acceleration and no brake. Manually collected fractions were screened by negative stain electron microscopy and those containing complexes were concentrated slowly and buffer exchanged to remove sucrose and glutaraldehyde using Zeba Spin desalting columns (Thermo Fisher Scientific).

### Negative stain EM

Though the concentration after purification was too low to quantify, samples were screened by negative stain electron microscopy for suitability for further structural studies. 2.5 μL of either C5b8-CD59, or C5b9-CD59 complexes were applied to carbon-coated copper grids (Agar Scientific) glow discharged in air. Grids were washed twice with water and then stained with 2% uranyl-acetate and were left to dry. Samples were imaged at a nominal magnification of 50k on a 120 kV Tecnai T12 microscope (Thermo Fisher Scientific) equipped with a 2 K Eagle CCD camera (Thermo Fisher Scientific).

### CryoEM grid preparation and data collection

C5b8-CD59 and C5b9-CD59 complexes were imaged on graphene oxide coated grids which were prepared in house[51]. Briefly, R1.2/1.3 Cu Quantifoil grids (Agar Scientific) were plasma cleaned for 1 minute, then 3 μL of 0.2 mg/mL graphene oxide solution (Sigma) was applied to the carbon side of the grid. Excess liquid was removed by blotting from the back. Graphene oxide was applied 3 times in total and the grids washed with water between each application. The grids were then washed a further two times with water and left to dry for 30 minutes. To vitrify grids, 2.5 μL of C5b8-CD59 or C5b9-CD59 complexes at a concentration 3 times greater relative to the negative stain experiments were applied to the graphene oxide side of the grids. After a wait time of 30 seconds, the sample was blotted for 2.5 seconds at a 'blot force' of −3 and plunge frozen in liquid ethane using a Vitrobot mark IV (Thermo Fisher Scientific). Electron micrograph movies were collected on a 300 keV Titan Krios (Thermo Fisher Scientific) fitted with a Falcon IV (C5b8-CD59 dataset) or a K3 (Gatan) (C5b9-CD59 datasets) direct electron detector. Datasets were collected via EPU 1.12.079 using aberration-free image shift with fringe-free illumination. C5b8-CD59 and three C5b9-CD59 datasets (D$_1$, D$_2$ and D$_4$) were collected in super-resolution mode and binned by 2 on camera, an additional dataset (D$_3$) was collected in super-resolution mode and binned during motion correction. Specific details for all five data collections are summarized in the Supplementary Table 1.

### Image processing

**C5b8-CD59.** Raw micrograph movies were corrected for beam-induced motion using the RELION implementation of Motioncorr2[52,53]. CTF parameters of motion-corrected micrographs were estimated using Gctf v1.06[54]. Particles (1,138,825) of C5b8-CD59 were picked using the general model in crYOLO v1.7.6[55] and coordinates were imported into RELION v3.1[53]. Particles were down sampled to 3.5 Å/pixel and extracted from micrographs using a box size of 128 pixel[2]. Down-sampled particles were imported into cryoSPARC v3.1.1[56] and subjected to two rounds of reference-free 2D classification. Particles from featureless, noisy, or poorly resolved classes were discarded. Particles (301,591) from well-resolved 2D classes were imported back into RELION using the csparc2star.py Python 3.6.5 script[57]. A reference model was generated using ab initio reconstruction in cryoSPARC. Particles were refined to 7.0 Å in RELION using the ab initio reference model low-pass filtered to 40 Å and subjected to 3D classification (4 classes, 50 iterations, 350 Å soft circular mask) in RELION. Two of the 3D classes showed a well-resolved C5b8-CD59 complex. Particles (206,782) from the two good classes were re-extracted at the original pixel size (1.171 Å/pixel) using a box size of 448 pixel[2]. Unbinned particles were refined to 3.4 Å, then subjected to Bayesian polishing and CTF refinement in RELION[58]. Polished particles were refined to 3.1 Å, imported into cryoSPARC and subjected to five iterations of global CTF refinement and a single round of local CTF refinement. Particles were then refined to a resolution of 3.0 Å. Local resolution was calculated, and a locally filtered map was generated in cryoSPARC using an ad hoc B-factor of −60 Å$^2$.

To better resolve density for CD59, we performed particle subtraction in RELION. A soft mask was created to encompass C8β, C8α, and CD59. Density outside the mask (C5b, C6, C7 Factor I-like Modules, and C8γ) was subtracted from the final set of 206,782 particles. To generate an initial model, the subtracted particles were manually

reconstructed using relion_reconstruct. The subtracted particles were imported into cryoSPARC v3.1.1 and refined to 3.1 Å. Subtracted particles were subjected to 5 iterations of global CTF refinement, a single round of local CTF refinement, and refined to a resolution of 2.9 Å. For the density subtracted map, local resolution was calculated and a locally filtered map was generated in cryoSPARC using an ad hoc B-factor of −60 Å². 3D Variability analysis of the density subtracted map was performed in cryoSPARC[59] with a filter resolution of 4 Å. The results were output using the 'simple' mode to generate 20 frames across the first principal component. Resolutions were determined using the gold-standard masked Fourier Shell Correlation (FSC) implemented in cryoSPARC.

**C5b9-CD59**. As with the C5b8-CD59 dataset, individual movie frames were motion corrected using the RELION implementation of Motion-Corr2. CTF parameters of motion corrected movies were estimated using the CTFFIND4 v4.1[60] wrapper in RELION 3.1. Particles were either picked either using a crYOLO model trained on 50 micrographs and imported into cryoSPARC v2 or picked directly with cryoSPARC v2 blob picking. Each dataset was handled separately. Particles were down sampled by a factor of 4 and subject to multiple rounds of 2D classification. As with C5b8-CD59 processing, particles from featureless, noisy, or poorly resolved classes were discarded. An initial model was generated in cryoSPARC, low pass filtered to 40 Å and used as a reference for a consensus refinement. Particles from each dataset were extracted at the original pixel size, imported into RELION 3.1 using csparc2star.py, and subjected to auto refinement in preparation for merging. Particles (974,249) from all four datasets were merged maintaining unique optics groups, subjected to Bayesian polishing followed by CTF refinement, and auto refined to generate a consensus map.

CryoDRGN was used to calculate an overall conformational landscape for the C5b9-CD59 complex[61]. To separate heterogeneity of C9 within the complex, we used extensive 3D classification methods implemented within RELION. First, we used multiple rounds of 3D classification with local searches to separate uninhibited complexes (round one) and obtain a single class (282,840 particles) with clear CD59 density (round two). Next, we implemented a refinement strategy using density subtraction and focused classification. Density corresponding to C5b6 and C7 was subtracted; the remaining arc was subjected to classification without alignment. Particles were reextracted and classes with the same numbers of C9 were merged, duplicates removed and subject to individual auto refine. We obtained classes with either 2 (62,128 particles), 3 (105,227 particles) or 4 (84,635 particles) C9 monomers. Next, we used a series of focused classification procedures to further separate our data. For complexes with 2 or 3 C9 molecules, we generated a spherical mask around CD59 and used 3D classification without alignment. For each of these complexes, the class with the best resolved CD59 density was subject to a round of auto refinement. The focused classification of the 4 C9 complex indicated a mixture of states. We separated mis-classified particles belonging to the 3 C9 complex and merged them with the previous class, removing duplicates. Together this led to the generation of two maps C5b9$_2$-CD59 (47,244 particles) at a global resolution of 3.3 Å and C5b9$_3$-CD59 (65,125 particles) at a global resolution of 3.3 Å.

To further improve the local resolution for CD59 in both maps, we subtracted density corresponding to C5b6, and C7. Local refinement of the remaining density improved the resolution for CD59 such that the central β-strand was unambiguously resolved (C5b9$_2$-CD59, 3.4 Å; C5b9$_3$-CD59, 3.1 Å). Local resolution was calculated, and locally filtered maps were generated in RELION using an ad hoc B-factor of −50 Å².

## Model building
Models were built and refined into the locally filtered maps using sMAC (PDB: 7NYD)[28] as a starting model. For C5b8-CD59, coordinates for all C9 molecules were removed. For the C5b9$_2$-CD59 complex, the sMAC model was trimmed back to include only domains for which there was well ordered density. Where density allowed, the glycans built for sMAC were maintained and the unfurled β-strands of the complement proteins were manually extended. Next, the crystal structure of CD59 (PDB: 2J8B)[29] was rigid body fit into the density next to the C8α hairpin. This model was refined in the density subtracted maps using ISOLDE v1.1.0[62] with secondary structure restraints and adaptive distance restraints. The C5b9$_3$-CD59 model was derived from the C5b9$_2$ model in which coordinates for the first C9 were duplicated and fit into the second position. Subsequently, the final C9 in the C5b9$_2$ model was shifted to the terminal position and the final C5b9$_3$-CD59 model was refined as described above. To generate models of complete complexes, we refined our models into the maps prior to density subtraction. Complete models were subjected to real-space refinement in Phenix v1.2[63], where secondary structure restraints and reference-based model restraints were maintained. Model B-factors were refined in Phenix. Map-model FSC and full cryoEM validation were assessed using inbuilt validation tools in Phenix v1.2[64]. The model for the murine CD8α-CD59 interface was generated using the AlphaFold2[65] prediction of CD59 (AlphaFold ID: O55186) and template-based model building of C8α using the online portal for SWISSMODEL[66].

## Map visualization and figure generation
All density maps, models and electrostatic surface representations were visualized in UCSF ChimeraX v1.1[67]. Local resolution maps and angular distributions were generated in RELION 3.1 and CryoSPARC. All map, model, and molecular dynamics figures were generated in ChimeraX.

## Generating fluorescently labeled reagents
C9 (CompTech) was chemically labeled with AlexaFluor-568 labeling kit (Thermo Fisher Scientific) as per manufactures instructions. Briefly, 250 µL of 1 mg/mL C9 was mixed with 50 µL of 1 M sodium bicarbonate. The mixture was then added to 1 vial of Alexafluor-568 reagent and stirred at room temperature for 1 h. The labeled protein was purified from the free label by spinning at 1000 x $g$ for 2 minutes through a Zeba Dye removal column (Thermo Fisher Scientific). The resin was then washed with 2 × 250 µL of phosphate buffered saline (PBS) to recover the C9 conjugate. Fractions were combined, aliquoted, flash frozen and stored at −80 °C for future use.

To generate cells expressing fluorescently labeled CD59, we transfected SNAP-tagged CD59 into Chinese hamster ovary-K1 (CHO-K1) cells (a gift from A. Kusumi). Cells were cultured in Hams-F12 media (Thermo Fisher Scientific) with 10% fetal bovine serum (FBS). Approximately 250,000 cells were seeded into 6 well plates and allowed to adhere overnight. Cells were transfected with 4 mg of SNAP-CD59 (Addgene) using Lipofectamine 3000 (Life Technologies). After 24 hours cells were detached with 1 mL of PBS supplemented with 1 mM EDTA. Cells were replated on poly-L-lysine coated 8-well chamber slides at 50,000 cells per well and allowed to adhere overnight. To stain for CD59, a 1 in 200 dilution of SNAP-Oregon cell permeable ligand (NEB) was added.

## Cholesterol depletion assays
Cholesterol depletion was carried out using 10 mM methyl-β-cyclodextrin in serum free media for 30 minutes at 37 °C. The extend of cholesterol depletion was assessed using an Amplex Red Cholesterol Assay (Thermo Fisher Scientific) as per manufacture instructions. Cells were washed and treated with 0.05 mg/mL of rabbit anti-CHO polyclonal IgG (catalog no: 27803-1-AP Proteintech) (80x dilution from the stock concentration) and incubated at room temperature for 20 minutes. Cells were washed twice with PBS to remove unbound antibody and treated with 5% C9 depleted serum (Sigma) supplemented with an excess of chemically labeled C9-Alexafluor 568

(Thermo Fisher Scientific). Cells were treated for 15 minutes at 37 °C then washed twice to remove complement proteins. Cells were fixed and mounted with DAPI nuclear stain. The slides were imaged on a Leica SP8 inverted confocal microscope at 60 x oil immersion objective lens, to visualize the distribution of C9 and SNAP-tagged CD59. Data was collected using the Leica Application Suite X software and analyzed using the Fiji-2 image analysis platform.

## Molecular dynamics simulations

**Atomistic simulations of GPI-anchored CD59.** Atomistic GPI-anchored was built using the Membrane Builder utility[68] of CHARMM-GUI[69-71] by submitting the structure of CD59 (PDB: 1CDR)[72] to the webserver and adding the GPI core with an extra phosphoethanolamine modification on the second mannose residue. The protein was placed within a DOPC bilayer in a 12 × 12 x 13.8 nm box, solvated with TIP3P water[73] and 150 mM NaCl (with two extra Na$^+$ ions for neutralization). All simulations were performed with Gromacs 2021.3[74]. Three independent replicates were run with the CHARMM36m forcefield[75]. The system was minimized with the steepest descent algorithm and a maximum force of 1000 kJ/mol before a 250 ps NVT equilibration at 310 K (1 fs timestep) using position restraints on the protein and lipids, followed by a 1 ns in a constant number of atoms, pressure, and temperature (NPT) equilibration (2 fs timestep) with backbone restraints only. The production runs were performed for 500 ns at 310 K with a Nose-Hoover thermostat (τt = 1.0 ps) and a with a pressure of 1 bar maintained by a semi-isotropic Parrinello-Rahman barostat (τp = 5.0 ps). The LINCS algorithm[76] was used to constrain bonds to hydrogen atoms.

**Coarse-grained (CG) simulations of C5b8-CD59.** The C5b8-CD59 model was prepared for MD simulations as follows: C5b was split into two independent chains to account for the chain break in C5b. Missing regions of the structure were rebuilt to reflect the full inserted conformations of C6, C7, and C8β using the coordinates from the full MAC open structure (PDB: 6H03)[30]. The TMHs of C8α were connected by a triple glycine linker and renumbered according. All glycans were removed from the model. The resulting model was converted to CG representation with the martinize.py script (Python 3.8.8) using the ELNEDYN framework[77] for MARTINI 2.2[78,79] with an elastic bond force constant of 500 kJ/mol/nm$^2$, and upper and lower elastic bonds cut-offs of 1.0 nm and 0.5 nm, respectively.

To model the native GPI anchor, we used published parameters for the sugars and lipid tails[80] and parameterized the linker specifically for CD59 using the same strategy outlined in[80]. An atomistic molecule was built with LEaP in Ambertools21[81] comprising of the three C-terminal residues of CD59 (Leu-Glu-Asn) with an acetyl cap on the N-terminus of the leucine residue. The C-terminal carbon of the asparagine was covalently bonded to the phosphoethanolamine (EtNP) linker, which was linked to the first two mannose moieties of the GPI anchor (Supplementary Fig. 9A). The GLYCAM06h forcefield[82] was used to build the molecule, which was placed in a box with 2,466 TIP3P water molecules and 2 neutralizing Na$^+$ ions. The system was converted for use in GROMACS 2021.3 with ACPYPE[83]. Energy minimization was performed for 1,000,000 steps or until convergence using the steepest descent algorithm, followed by NPT equilibration for 100 ps and a 200 ns run, both using the sd integrator[84]. The same molecule was converted to a CG representation in which the EtNP linker corresponds to two beads: L1 and L2 (Supplementary Fig. 9B). The CG molecule was placed in a 5 × 5 x 5 nm$^3$ box filled with polarizable water and two neutralizing sodium ions, using a modified version of the Martini 2.2 forcefield[79,85]. The system was minimized for 1,000,000 steps or until convergence using the steepest descent algorithm before a 500 ps NPT equilibration and a 200 ns run with the sd integrator. The bead parameters were iteratively adjusted to match the behavior of the atomistic molecule, as measured in three 200 ns simulations

(Supplementary Fig. 9). The final CG parameters are reported in Supplementary Table 2. The parameters for L1 and L2 were integrated into the C5b8-CD59 model.

The complex was placed in 7:3 (molar ratio) DOPC:cholesterol membrane in a 22.5 × 22.5 × 30 nm box using the insane.py script (Python 2.7.5)[86], with polarizable waters[87] and 150 mM of neutralizing NaCl. Three replicates were run independently in GROMACS 2021.3. For each replicate, the system was first minimized with the steepest descent algorithm with a maximum force of 100 kJ/mol. The minimized system was then equilibrated for 2 ns (1 fs timestep) NPT ensemble with a pressure of 1 bar kept by a semi-isotropic Berendsen thermostat (τp = 5.0 ps). The equilibrated system was then subjected to a 2 μs (5 fs timestep) production run in an NPT ensemble maintained at 1 bar by a semi-isotropic Parinello-Rahman barostat (τp = 12.0 ps). For both the equilibration and production runs, the leapfrog stochastic dynamics integrator (sd)[84] was used and the temperature was kept at 310 K with the v-rescale thermostat (τt = 1.0 ps). The relatively small timesteps were necessary to avoid numerical instabilities, and the Lennard-Jones potentials describing sugar-sugar and sugar-protein interactions were scaled down by a factor of 0.85 to prevent aggregation[80]. The Verlet scheme was used to impose a cut-off of 1.1 nm for both van der Waals (vdW) and Coulomb. A smooth particle mesh Ewald method[88] was used was used for electrostatics and the plain cut-off method was used for vdW interactions. The relative dielectric constant was set at 2.5, which is the default value for polarizable water in MARTINI 2.2.

## Reporting summary

Further information on research design is available in the Nature Portfolio Reporting Summary linked to this article.

## Data availability

Source data are provided with this paper. Raw data generated in this study underlying Supplementary Fig. 1B and Supplementary Fig. 8E are included as a Source Data file. Initial and final configurations for MD simulations are included in Supplementary Data Files 1–12. The cryoEM maps generated in this study have been deposited in the Electron Microscopy Data Bank under the accession codes EMD-15779 (C5b8-CD59), EMD-15781 (C5b9$_2$-CD59), and EMD-15780 (C5b9$_3$-CD59). The structural models generated in this study have been deposited in the Protein Data Bank under the accession codes 8H0F (C5b8-CD59), 8B0H (C5b9$_2$-CD59), and 8B0G (C5b9$_3$-CD59). Structural models used to initiate model building were accessed from the Protein Data Bank under the accession codes 7NYD, 2J8B and 6H03. Structural models used in data analysis were accessed from the Protein Data Bank under the accession codes 5IMT, 7NYD, 6H03, and from the AlphaFold protein structure database entry O5518. Structural models used to generate Fig. 1 were accessed from the Protein Data Bank under the accession codes 6H03, 3T5O, 3OJY, 6CXO, and from the AlphaFold protein structure database entry P10643. Source data are provided with this paper.

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

## Acknowledgements

We thank R. Smith for gifting CD59; A. Kusumi for CHO-K1 cells; S. Islam for computational support; P. Simpson and N. Cronin for EM support. Initial screening of samples was carried out at Imperial College London Centre for Structural Biology; cryoEM data used to calculate the final reconstructions was collected at Diamond Light Source and London Consortium for CryoEM (LonCEM) (Francis Crick Institute, UK). Fluorescence microscopy images were collected at the FILM facility (Imperial College) and the Advanced Light Microscopy STP (Francis Crick Institute, UK). This project has received funding from the European Research Council (ERC) under the European Union's Horizon 2020 research and innovation programme (grant agreement No. 864751) to DB; ECC. is

supported by the CRUK Convergence Science Centre at Imperial College London (C24523/A26234): ECC and JKB are supported by the EPSRC Centre for Doctoral Training: Chemical Biology: Physical Sciences Innovation (EP/L015498/1). TBV. is funded by BBSRC Doctoral Training Program grant (BB/M011178/1). Research in PJS's lab is funded by Wellcome (208361/Z/17/Z), the MRC (MR/S009213/1) and BBSRC (BB/P01948X/1, BB/R002517/1 and BB/S003339/1). We thank Diamond for access and support of the Cryo-EM facilities at the UK national electron bio-imaging centre (eBIC), proposal BI25127, funded by the Wellcome Trust, MRC and BBSRC. This project made use of time on HPC granted via the UK High-End Computing Consortium for Biomolecular Simulation, HECBioSim (http://hecbiosim.ac.uk), supported by EPSRC (grant no. EP/R029407/1) to HECBioSim.

## Author contributions

Conceptualization: DB; Methodology: E.C.C., S.G., T.B.V., J.K.B.; Investigation: E.C.C., S.G., T.B.V.; Funding acquisition: D.B., E.W.T.; Project administration: D.B., E.W.T., P.J.S.; Supervision D.B., E.W.T., P.J.S.; Writing-original draft: D.B.; Writing review and editing: D.B., E.C.C., S.G., T.B.V., E.W.T. and P.J.S.

## Competing interests

EWT is a founder and shareholder of Myricx Pharma Ltd. All other authors declare no competing interests.
