## [Peer review file · Nature Communications]

REVIEWER COMMENTS

Reviewer #1 (Remarks to the Author):

Highly significant and original paper revealing how complement regulatory protein CD59 prevents membrane attack formation. Through several cryoEM structures (of both C5b8 and C5b-9 in complex with CD59) authors reveal the mechanism of CD59 inhibition (which was long elusive). In addition, the influence of the membrane environment on MAC assembly and inhibition is explained.

Excellent work that is of great importance to the immunology field. Exact mechanism of CD59 inhibition was not yet known.

Methodology is sound and interpretations are correct.

minor comment: worth mentioning in the introduction is a recent paper by Shao et al showing a critical role of CD59 upregulation in cancer development. Nature Cancer 3, 1192–1210 (2022).
<https://doi.org/10.1038/s43018-022-00444-4>

Reviewer #2 (Remarks to the Author):

The authors present a combined experimental and simulation-based study on the action of CD59. They convincingly show using cryo-EM how CD59 shapes and thereby regulates MAC, clearly an important, interesting, and timely study.

They further show that MAC assembly is clustered on the plasma membrane. The authors claim that these domains are cholesterol-rich lipid rafts. This is a very popular claim, there is a huge tendency to interpret results in terms of rafts. However, methyl-beta-cyclodextrin as used here may have pretty dramatic effects on the cell membrane, it not only extracts cholesterol but as well other lipids. Is there any additional support for rafts, e.g. do the performed simulations show any enhanced cholesterol

content close to CD59? What was the reason for placing CD59 in a pure DOPC bilayer in the atomistic MD simulation? This is surprising, considering that DOPC yields a disordered, fluid-phase membrane while the authors expect CD59 to reside in ordered, cholesterol-rich raft domains.

Minor points:

- It would be great if the authors could add a figure to the introduction that sketches the MAC-related processes described.

- The authors conclude from simulation data and experiment a local distortion of the lipid bilayer to 'alter the physical properties..'. What is the meaning of this very local thinning for function, how are the physical properties changed?

- The number of minimization steps in the setup of the coarse-grained simulations appears very high. Was this really required?

- It would be helpful if the authors could add some more information on the simulation parameters used (vdw, electrostatics, cut-off..)

Reviewer #3 (Remarks to the Author):

The manuscript by Couves et al. uses combination of single particle cryo-EM cellular assays, and molecular dynamics to explain how the CD59 binds complement proteins and inhibits the formation of the membrane attack complex (MAC). The MAC is a human immune pore that is formed by oligomerization of complement proteins. It is an important part of innate immunity – however, if oligomerization and pore formation is not regulated correctly it can cause serious damage to human cells. This manuscript is a careful study of using of single particle cryo-EM to show how the relatively “tiny” CD59 human cell surface receptor that inhibits MAC pore formation, but also is a target of bacterial virulence factors. The authors took the approach of anchoring CD59 to nanodiscs and then using single particle cryo-EM and image processing approaches to determine many structures that show various features of how CD59 inhibits MAC formation. They test these models using cellular assays and molecular modeling.

This is a careful and thorough structural study, a true tour-de-force. The authors should be commended on the quality of their structures and the transparent and easy to follow presentation of their data processing steps for how the structures were determined and how different image processing approaches were used to get the most out of their data. While my expertise is not in the cell biology or the molecular dynamics, but these experiments added additional data that tested some of the models predicted by their structures. This work was a pleasure to read and should be of interest to a broad range of biological and structural scientists.

Structural basis for membrane attack complex inhibition by CD59

Couves et al.

Response to reviewers

We are writing in response to your e-mail message with peer-review comments on our manuscript “Structural basis for membrane attack complex inhibition by CD59” (NCOMMS-22-42598-T). We were pleased and encouraged by the reviewers’ recognition of the interest and significance of our results. We thank the reviewers for their suggestions on how we could strengthen and improve the manuscript. We have now revised our manuscript to address the reviewers’ concerns, as detailed below.

Reviewer #1 (Remarks to the Author):

Highly significant and original paper revealing how complement regulatory protein CD59 prevents membrane attack formation. Through several cryoEM structures (of both C5b8 and C5b-9 in complex with CD59) authors reveal the mechanism of CD59 inhibition (which was long elusive). In addition, the influence of the membrane environment on MAC assembly and inhibition is explained.

Excellent work that is of great importance to the immunology field. Exact mechanism of CD59 inhibition was not yet known. Methodology is sound and interpretations are correct.

Response: We thank the reviewer for their recognition of our work.

Minor comment: worth mentioning in the introduction is a recent paper by Shao et al showing a critical role of CD59 upregulation in cancer development. Nature Cancer 3, 1192–1210 (2022). <https://doi.org/10.1038/s43018-022-00444-4>

We thank the reviewer for highlighting this important study published after our initial submission. This reference is now included in the introduction (reference number 6):

“CD59 is also highly expressed in B-cell lymphomas, contributing to immune evasion and protection from antibody-based therapeutics that activate complement^{5,6}.”

Shao, F. et al. Silencing EGFR-upregulated expression of CD55 and CD59 activates the complement system and sensitizes lung cancer to checkpoint blockade. Nat Cancer 3, 1192-1210 (2022).

Reviewer #2 (Remarks to the Author):

The authors present a combined experimental and simulation-based study on the action of CD59. They convincingly show using cryo-EM how CD59 shapes and thereby regulates MAC, clearly an important, interesting, and timely study.

They further show that MAC assembly is clustered on the plasma membrane. The authors claim that these domains are cholesterol-rich lipid rafts. This is a very popular claim, there is a huge tendency to interpret results in terms of rafts. However, methyl-beta-cyclodextrin as used here may have pretty dramatic effects on the cell membrane, it not only extracts cholesterol but as well other lipids. Is there any additional support for rafts e.g. do the performed simulations show any enhanced cholesterol content close to CD59?

Response: Although our simulations do not show an enhancement of cholesterol near CD59, previous studies have shown that antibody-based therapeutics that activate complement target lipid rafts, either because lipid rafts favour complement activation or because rafts are more sensitive to MAC-mediated lysis. This is now reflected in the text with the corresponding reference cited:

“These data are consistent with assays showing monoclonal antibody therapeutics that activate complement target membrane rafts³⁶.”

Cragg, M.S. et al. Complement-mediated lysis by anti-CD20 mAb correlates with segregation into lipid rafts. *Blood* **101**, 1045-52 (2003).

What was the reason for placing CD59 in a pure DOPC bilayer in the atomistic MD simulation? This is surprising, considering that DOPC yields a disordered, fluid-phase membrane while the authors expect CD59 to reside in ordered, cholesterol-rich raft domains.

Response: The goal of the atomistic simulation of CD59 was to verify that CD59 could adopt a range of conformations relative to the membrane, including the orientation observed in our cryoEM maps. DOPC was chosen in the atomistic MD simulation of CD59 as it was the lipid composition used in our cryoEM structure.

Minor points:

- It would be great if the authors could add a figure to the introduction that sketches the MAC-related processes described.

Response: This is now done and is referred to as Supplementary Fig. 1 the revised manuscript.

- The authors conclude from simulation data and experiment a local distortion of the lipid bilayer to 'alter the physical properties..'. What is the meaning of this very local thinning for function, how are the physical properties changed?

Response:

Our previous work has shown that physical properties of the membrane, such as bending modulus, are altered during MAC assembly (Menny et al., *Nature Communications* 2018). While the precise impact of membrane thinning on MAC function is yet to be explored, hydrophobic mismatches between proteins and the lipid bilayer can influence membrane protein interactions and diffusion (Jiang et al., *Nature Communications* 2022). We have now added a sentence in the discussion to reflect this.

“While the implications of this thinning on MAC function remains to be explored, hydrophobic mismatches between proteins and the lipid bilayer can influence membrane protein interactions and diffusion³⁹.”

Jiang, Y. et al. Membrane-mediated protein interactions drive membrane protein organization. *Nat Commun* **13**, 7373 (2022).

- The number of minimization steps in the setup of the coarse-grained simulations appears very high. Was this really required?

Response: We thank the reviewer for their careful reading of the manuscript. Indeed, this was an error. The correct number of minimization steps is 1,000,000. We recognize that this number of steps is still higher than what would be expected for a nonGPI-anchored protein. The publication (Banerjee

et al., 2020) describes the first and only instance of a GPI anchor parameterized for MARTINI2. They show that likely due to the coarse-grain sugars of the GPI anchor, these simulations are prone to numerical instabilities. In our GPI-anchored system, we also observe these instabilities and have modified the methods section to clarify that minimizations were performed for 1,000,000 steps or until convergence.

“Energy minimization was performed for 1,000,000 steps or until convergence using the steepest descent algorithm”

- It would be helpful if the authors could add some more information on the simulation parameters used (vdw, electrostatics, cut-off..)

Response: We now included these details in the methods section.

“The Verlet scheme was used to impose a cut-off of 1.1 nm for both van der Waals (vdW) and Coulomb. A smooth particle mesh Ewald method was used for electrostatics and the plain cut-off method was used for vdW interactions. The relative dielectric constant was set at 2.5, which is the default value for polarizable water in MARTINI 2.”

Reviewer #3 (Remarks to the Author):

The manuscript by Couves et al. uses combination of single particle cryo-EM cellular assays, and molecular dynamics to explain how the CD59 binds complement proteins and inhibits the formation of the membrane attach complex (MAC). The MAC is a human immune pore that is formed by oligomerization of complement proteins. It is an important part of innate immunity – however, if oligomerization and pore formation is not regulated correctly it can cause serious damage to human cells. This manuscript is a careful study of using of single particle cryo-EM to show how the relatively “tiny” CD59 human cell surface receptor that inhibits MAC pore formation, but also is a target of bacterial virulence factors. The authors took the approach of anchoring CD59 to nanodiscs and then using single particle cryo-EM and image processing approaches to determine many structures that show various features of how CD59 inhibits MAC formation. They test these models using cellular assays and molecular modeling.

This is a careful and thorough structural study, a true tour-de-force. The authors should be commended on the quality of their structures and the transparent and easy to follow presentation of their data processing steps for how the structures were determined and how different image processing approaches were used to get the most out of their data. While my expertise is not in the cell biology or the molecular dynamics, but these experiments added additional data that tested some of the models predicted by their structures. This work was a pleasure to read and should be of interest to a broad range of biological and structural scientists.

Response: We thank the reviewer for their positive comments.